# Novel BH4-BCL-2 Domain Antagonists Induce BCL-2-Mediated Apoptosis in Triple-Negative Breast Cancer

**DOI:** 10.3390/cancers14215241

**Published:** 2022-10-26

**Authors:** Vishnupriya Kanakaveti, Sakthivel Ramasamy, Rahul Kanumuri, Vaishnavi Balasubramanian, Roshni Saravanan, Inemai Ezhil, Ravishankar Pitani, Ganesh Venkatraman, Suresh Kumar Rayala, M. Michael Gromiha

**Affiliations:** 1Department of Biotechnology, Bhupat and Jyoti Mehta School of Biosciences, Indian Institute of Technology Madras, Chennai 600036, Tamil Nadu, India; 2Division of Oncology, Departments of Medicine and Pathology, Stanford University School of Medicine, Stanford, CA 94305, USA; 3Department of Human Genetics, Sri Ramachandra Faculty of Biomedical Sciences & Technology, Sri Ramachandra Institute of Higher Education & Research, Porur, Chennai 600116, Tamil Nadu, India; 4Department of Community Medicine, Sri Ramachandra Medical College, Sri Ramachandra Institute of Higher Education & Research, Porur, Chennai 600116, Tamil Nadu, India

**Keywords:** triple-negative breast cancer, BH4 domain, covalent BH4 inhibitors, targeted therapy

## Abstract

**Simple Summary:**

Our work has led to the identification of three novel BH4 mimetics, SM216, SM396, and SM949, with nanomolar activities both *in vitro* and *in vivo* assays. SM396 binds covalently to the BH4 domain of BCL-2 while the compounds SM216 and SM949 are non-covalent BH4 binders. Our results illustrate that these compounds are highly specific to the triple-negative breast cancer cells with no effect on normal cells. Elevated levels of Cyt-c induced by these compounds suggest significant inhibition of BCL-2 leading to apoptosis. Further investigations of these potent lead compounds will lead to clinical translations in targeting challenging tumor types.

**Abstract:**

Targeting the challenging tumors lacking explicit markers and predictors for chemosensitivity is one of the major impediments of the current cancer armamentarium. Triple-negative breast cancer (TNBC) is an aggressive and challenging molecular subtype of breast cancer, which needs astute strategies to achieve clinical success. The pro-survival B-cell lymphoma 2 (BCL-2) overexpression reported in TNBC plays a central role in deterring apoptosis and is a promising target. Here, we propose three novel BH4 mimetic small molecules, SM396, a covalent binder, and two non-covalent binders, i.e., SM216 and SM949, which show high binding affinity (nM) and selectivity, designed by remodeling the existing BCL-2 chemical space. Our mechanistic studies validate the selectivity of the compounds towards cancerous cells and not on normal cells. A series of functional assays illustrated BCL-2-mediated apoptosis in the tumor cells as a potent anti-cancerous mechanism. Moreover, the compounds exhibited efficacious *in vivo* activity as single agents in the MDA-MB-231 xenograft model (at nanomolar dosage). Overall, these findings depict SM216, SM396, and SM949 as promising leads, pointing to the clinical translation of these compounds in targeting triple-negative breast cancer.

## 1. Introduction

Evading apoptosis is a key hallmark of cancer, owing to the aberrant expression of pro-survival BCL-2 proteins governing the intrinsic apoptotic pathway [1]. The fate of the cancer cells exposed to the chemotherapeutic agents and the commitment to death depends on BCL-2 and its interactions [2]. Restraining the binding of pro-survival proteins (BCL-2, BCL-xL, BCL-W, A1, and MCL-1) and pro-apoptotic proteins (Sensitizers (BH3 only proteins) and effectors (BAX and BAK)) by BH3 mimetics induce cell death [3,4]. The N-terminal amphipathic BH4 (aa6-31) domain is a key domain to perform the anti-apoptotic function interacting with multiple partners (BAX, RAS, PP2A, and CED-4). The survival activity of BCL-2 family members is lost by either its cleavage or removal upon mutations [5,6]. The BH4 domain is capable of interacting with the pro-apoptotic BAX and also engages with non-BCL-2 family members, such as c-MYC Ras, voltage-dependent anion channel (VDAC), apoptosis stimulating of p53 protein 2 (ASPP2), inositol 1,4,5-triphosphate receptor (IP3R), cell death protein 4 (CED-4), calcineurin. and other important players. Thus, it plays a key role in various processes namely, Ca^2+^ ion uptake, DNA damage response, and genetic instability promoting tumorigenesis [7]. Targeting the BH4 domain with small molecules would be ideal in intervening in the BCL-2 anti-apoptotic function in various resistant cancers.

Triple-negative breast cancer is an aggressive and heterogenous molecular subtype of breast cancer with significant clinical challenges, lacking potential markers and predictors of better prognosis [8,9,10,11,12]. Until now, regimens are confined to anthracyclines, platinum derivatives, and taxanes as a mainstay of chemotherapy with limited clinical success [13,14,15,16]. Recently, BCL-2 has emerged as a promising prognostic marker in TNBC patients and its expression is associated with chemoresistance to anthracycline (doxorubicin)-based therapies [17,18]. Various reports demonstrated the sensitizing effect of BCL-2 inhibition with small molecule inhibitors viz. ABT-737, ABT-263 in breast cancer cell lines [19,20], and triple-negative breast tumor xenografts in combination with other cytotoxic agents. Unlike these small molecules, which induce thrombocytopenia as an off-target effect, venetoclax (ABT-199) is a platelet-sparing BCL-2 selective inhibitor [21,22]. Though ABT-199 was shown to elicit better responses, the latent resistant mechanisms [23,24,25,26,27] have sparked the need for scaffold optimization or a novel scaffold to achieve the targeted BCL-2 inhibition. Owing to this fact, our focus was exclusively on TNBC, which lacks therapeutic targets.

We observed a vacuum in the existing tactics in achieving high affinity as well addressing chemoresistance. To solve this problem, we propose a focused library of scaffolds (from insights of key functional groups from our previous computational model) for identifying selective and pan BCL-2 inhibitors [28]. Our method includes molecular remodeling of seven existing scaffolds using molecular hopping and hierarchical virtual screening concepts [29]. This resulted in 29 novel compounds, of which, three compounds (SM216, SM396, and SM949) outperformed the existing ones in terms of efficacy and drug-likeness. We then systematically investigated the *in vitro* and *in vivo* potency of the compounds selectively, targeting BCL-2 in TNBC cells. Further, the effectiveness of the compounds only on cancer cells with minimal effects on normal cell types made them ideal drug candidates. Remarkably, SM396 is likely to be the first covalent BCL-2 selective inhibitor binding to the BH4 domain; SM216 and SM949 are non-covalent inhibitors targeting TNBC cells as single agents. 

## 2. Methods

### 2.1. Compounds

The 29 lead molecules and ABT-199 were purchased from TIMTEC LLC and Selleck-chemicals, LLC, respectively. Compounds were dissolved in DMSO (cat no. D8418, Molecular Biology grade, Merck, Ltd., Kenilworth, NJ, USA) and were utilized for all cellular and functional assays.

### 2.2. Covalent Docking

The ligands were prepared and energy was minimized using the OpenBabel MMFF94 forcefield program. The flexible chain method was employed, which was defined for nitrogen and oxygen atoms of sidechains of tyrosine and lysine. 

We selected the structures of BCL-2 proteins (PDB codes: 6O0K, 6O0L, 6O0P, 6O0M, and 6O0O [30]) deposited in the Protein Data Bank [31] to estimate the affinity of the novel lead molecules. The docking simulations were performed using AutoDock 4.2 [32]. We considered the grid coordinates around the BH4 domain of BCL-2 (X, Y, Z coordinates of the center: −4.090, 6.298, 5.644; the number of points in X, Y, Z: 74, 80, and 78, respectively; spacing: 0.375Å) for the entire screening process [33]. The best docking results were considered based on the distance between O/N atoms, ≤2.0 Å for alkylating events, and the docking score was used to rank residue alkylation propensity.

### 2.3. Cell Culture

The cell lines A549, NCI-H460, NCI-H1299, HeLa, HCT-116, BT474, ZR-75, MCF7, and MDA-MB-231 were maintained in Dulbecco’s Eagle Medium (DMEM high glucose (4.5 g/L)) from either MP Bio or Hi Media supplemented with 10% fetal bovine serum (FBS) (HyClone Laboratories, Inc., Logan, UT, USA) and 100 U/mL penicillin and streptomycin antibiotic solution. The human mammary epithelial cells, HMEC (Lonza, Basel, Switzerland), were cultured in Eagle’s Minimal Essential Medium (MEM, Lonza) in accordance with the culturing instructions. The human pancreatic ductal epithelial cell line, HPDE (Lonza), was maintained in keratinocyte serum-free media (KSFM) as provided by the manufacturer. The human oral keratinocytes and skin keratinocytes were cultured in Dulbecco’s Eagle Medium (DMEM) with 4 mM of L-glutamine, 1.5 g/L of sodium bicarbonate, supplemented with 10% of FBS and 100 U/mL of penicillin and streptomycin solution. EAHy926 cells (a gift from Dr. Nitish Mahapatra, IITM) were cultured in endothelial growth medium with 10% FBS, 100 U/mL of penicillin and streptomycin antibiotic solution, and passaged using 0.25% of Trypsin-EDTA solution. All cell lines were maintained at 37 °C, 5% CO_2_ in a humidified incubator. 

### 2.4. Surface Plasmon Resonance 

Binding studies were performed using ProteOn XPR36 Protein Interaction Assay V.3.1 from Bio-Rad. Upon GLM chip activation with EDC (1-Ethyl-3-[3-dimethylaminopropyl] carbodiimide hydrochloride) and sulfo-NHS (*N*-hydroxysulfosuccinimide) (Sigma, St. Louis, MO, USA), the BCL-2 protein (Sigma) was immobilized using a 10 mM sodium acetate buffer pH 4.0 at 30 μL/min for 100 s in various channels except for the blank channel. The response units for immobilization were monitored until ~4000 RU followed by quenching the excess with 1 M of ethanolamine. At a flow rate of 20 μL/min, the compounds were passed onto the BCL-2 immobilized chip surface for 300 s followed by dissociation for 600 s. The kinetics of the association and dissociation were analyzed and fitted by a 1:1 Langmuir interaction model using ProteOn Manager. The time scale analysis was performed by pre-incubating the BCL-2 immobilized chip with SM396 and the kinetics were studied at 0 h, 3 h, and 6 h time points.

### 2.5. ESI Mass Spectrometry

BCL-2 protein (0.2 mg mL^−1^) was incubated overnight with 1 mM SM396 (prepared as stocks at 10 mM in 100% DMSO) only and then using C4 ZipTips (Millipore, Burlington, MA, USA) the samples were desalted according to the manufacturer’s instructions. The samples were dissolved in 1:1 (*v*/*v*) acetonitrile and water + 0.1% formic acid solvents and were analyzed at a flow rate of 20 μL/min by electrospray ionization (ESI) into a TOF mass spectrometer(Orbitrap), selecting the positive ion mode. The system was calibrated using myoglobin for the analysis. Using the PRIDE Inspector program, the resultant data were deconvoluted into mass spectra [34].

### 2.6. Immunoblotting

The cell lysates were subjected to protein extraction using a radio immunoprecipitation assay (RIPA) buffer (20 mM Tris-HCl (pH 7.5) 150 mM NaCl, 1 mM Na_2_EDTA, 1 mM EGTA, 2.5 mM sodium pyrophosphate, 1 mM Na_3_VO_4_ 1% sodium deoxycholate, 1 mM β-glycerophosphate, 1% NP40, and 1 µg/mL leupeptin). The protein content of the stored lysates (−80 °C) was quantified by the Bradford assay (Bio-Rad, Hercules, CA, USA). The protein samples were then denatured at 95 °C for 7–10 min, the samples were loaded onto the SDS-PAGE system followed by nitrocellulose transfer. Prior blocking of membranes in 5% skimmed milk in TBS and 0.1% Tween20 (blocking buffer) was performed before incubating with antibodies. Actin (Rabbit, no. 4967, CST) monoclonal antibody was used as the loading control. BCL-2 monoclonal antibody was purchased from CST (Rabbit, no. 4223).

### 2.7. BCL2 Gene Expression Analysis by qPCR

For extracting the total RNA, the cells were treated with TRIzol reagent (Thermo fisher scientific, Waltham, MA, USA). The cDNA was obtained by reverse transcription using the PrimeScript RT reagent kit (Takara Bio Inc, Shiga, Japan). The PCR-cycler (Agilent Technologies SureCycler 8800) was set at a temperature of 37 °C with a runtime of 15 min followed by heat inactivation at 85 °C for 5 sec and cooled down at 4 °C. The cDNA was diluted (2–10-fold) and subjected to conventional real-time PCR on an optical 96-well plate using the ABI 7500 Real-time PCR system (Takara Bio USA Inc, San Jose, CA, USA). The reagent volumes used were as follows: 30 μL of reaction mixture containing 2X SYBR Green master mix, 2.5 μL of assay mix, and 5 μL of cDNA. The wells with only the RNase-free water were considered as no template controls (NTCs). Actin was considered as the reference gene and the results were normalized to the levels of actin transcript. The relative expression was calculated as 2^−ΔΔCT^ using the formula ΔΔCT = ΔCT_Treated_ − ΔCT_Control_. The results were statistically evaluated using a *t-*test.

### 2.8. MTT Cell Viability Assay

An MTT (3-(4,5-dimethylthiazol-2-yl)-2,5-diphenyltetrazolium bromide)-based colorimetric assay was performed to measure the cell viability. On a 96-well plate, cells were seeded at a density of 3000–5000 cells/well. The cell number for the assay was decided based on the exponential growth of untreated cells. The cells were treated with compounds for 72 h followed by incubation with MTT for 4 h at 37 °C. The formazan crystals were dissolved in DMSO and absorbance was measured at 540 nm. The percentage viability was calculated as % cell viability = (OD treated cells/OD control cells) × 100, and the IC_50_ was estimated by curve-fitting using the non-linear regression of a dose–response curve. The experiments were repeated at least two to three times.

### 2.9. BrdU Cell Proliferation Assay

The effect of compounds on the proliferative capabilities of MDA-MB-231 cells was quantified using a BD Pharmingen BrdU flow kit (BD Pharmingen™, BD Biosciences, NJ, USA, Becton, Dickinson and company, NJ, USA). 

The assay was carried out according to the manufacturer’s protocol after 6 h treatment with compounds. Finally, the incorporated BrdU was measured using anti-BrdU APC, and the total DNA content with the 7-AAD stain using flow cytometric analysis.

### 2.10. Annexin V/Propidium Iodide Assay

Upon treating the MDA-MB-231 cells with lead compounds and incubating for 4 h, the cells were centrifuged and washed with a binding buffer (10 mM HEPES, 2.5 mM CaCl_2_, and 140 mM NaCl). After 15 min of incubation at room temperature with 200 µL of an Annexin-PI reagent (Annexin V-Alexa fluor 488 (Invitrogen, Waltham, MA, USA) and propidium iodide (Sigma)), the samples were loaded onto flow cytometer with 400 μL of a binding buffer at 4 °C. Each sample containing 10^4^ cells was analyzed by flow cytometry in an Epics XL/MCL flow cytometer (Beckman Coulter, Brea, CA, USA). The fluorescence was assessed at wavelengths of 520 nm (Alexa Fluor 488) and 630 nm (propidium iodide).

BT549 cells were treated with the DMSO, SM216, SM396, SM949, and ABT199 for 6 h. After treatment, the cells were trypsinized and washed with ice-cold PBS. The cell density was counted and 5 × 10^5^ cells were added to 500 μL of binding buffer (Invitrogen Alexa Flour 488 Annexin V Apoptosis kit, V13241). A total of 100 μL of the sample containing 10^5^ cells were incubated for 15 min at room temperature in the dark with 5 μL of Annexin V-Alexa Fluor 488 and 1 μL of 100 μg/mL propidium iodide (provided in the kit). Prior to flow cytometry analysis, an additional 300 μL of binding buffer was added. Each sample containing 10^5^ cells was analyzed in a BD Accuri FACS Canto II flow cytometer. Fluorescence for 10,000 events was collected at 520 nm (Alexa fluor 488) and 630 nm (propidium iodide).

### 2.11. Cellular BH4 Profiling Assay

The compounds SM216, SM396, SM949, and ABT-199 were arrayed onto a black, untreated 96-well plate as per the instructions provided [35] using an experimental buffer solution (150 mM Mannitol, 50 KCl, 20 mM EDTA, 20 mM EGTA, 0.1% BSA, 5 mM succinate, 10 mM Hepes-KOH, pH 7.5, 0.005% wt/vol digitonin, 10 mM 2-mercaptoethanol, 2 mM JC-1, and 20 mg/mL oligomycin). A total of 60 µL of each treatment was prepared for an equal proportion of MDA-MB-231 cells (1.67 × 10^6^ cells per milliliter), suspended in mitochondrial extraction buffer (150 mM Mannitol, 50 mM KCl, 20 mM EDTA, 20 mM EGTA, 0.1% BSA, 5 mM succinate, 10 mM Hepes-KOH, Ph 7.5). Control wells were arrayed with an alamethicin peptide and 20 mM of carbonyl cyanide-4-(trifluoromethoxy)phenylhydrazone (FCCP), a chemical uncoupler of oxidative phosphorylation. 

The final concentrations of the compounds were adjusted to 1 µM for treatment. The fluorescence of JC-1 aggregates was measured in kinetic mode at 5 min intervals for a period of 3 h at 590 nm, with an excitation wavelength at 545 nm on a SpectraMax5 (Molecular Devices) plate reader [36]. The area under the response curve was normalized to untreated and FCCP data to identify the percent depolarization. Curves were plotted in GraphPad Prism 6.

### 2.12. Cytochrome C Release Assays 

After 6 h of treatment with SM216, SM396, and SM949, MDA-MB-231 cells were washed with PBS followed by addition of Mannitol Extraction buffer, and incubated for 15 min. First, cells were centrifuged at 4 °C with RCF of 700× *g* for 10 min. The supernatants were collected and centrifuged at 4 °C for 30 min at RCF, 10,000× *g*. The cytosolic fractions (supernatant) were analyzed for the Cytochrome C release using the ELISA kit (Invitrogen). Finally, the Cytochrome C release was quantified and data were presented for two independent experiments.

### 2.13. Clonogenic Colony Formation Assay

MDA-MB-231 cells at 90% confluency were trypsinized and seeded (300 cells/plate) onto a 60 mm cell culture dish. Cells were cultured in Dulbecco’s Modified Eagle Medium (DMEM, Hg) supplemented with or without compounds SM216, SM396, and SM949, 10% fetal bovine serum, and 100 units/mL of penicillin and streptomycin solution maintained at 37 °C, 5% CO_2_. Cells were grown for 2 weeks until the colonies were formed. Colonies were after staining with 0.05% crystal violet and counted.

### 2.14. Soft Agar Colony Formation Assay

Soft agar was prepared (1.8% stock *w*/*v*, in distilled water) by dissolving and boiling the solution. The soft agar was aliquoted as the bottom layer (0.6% in growth medium) and allowed to solidify. MDA-MB-231 cells were trypsinized at 90% confluency and suspended in agar as the top layer (0.3% in growth media) at a density of 10^4^ cells/plate. Cells were maintained at 37 °C in a humidified chamber at 5% CO_2_. Cells were grown for 2–3 weeks with the intermittent supply of growth medium in the presence or absence of drugs and counted after staining with 0.05% crystal violet.

### 2.15. Wound Healing Assay

MDA-MB-231 and BT549 cells were seeded onto 35 mm culture dishes. At 95% confluency, the cells were serum-starved overnight, prior to scratching using a pipette tip (p20). The cells were washed twice with PBS after the scratch and treated with DMSO, SM216, SM396, SM949, and ABT199 for 24 h. Cell migration was monitored and images were captured at regular intervals (0 h, 12 h, and 24 h) using Leica Thunder Imaging System (LASX Application Suite). 

The widths of the wounds were measured for each image using ImageJ software and an average of 3 measurements was calculated.

The percentage migration of cells was evaluated using the formula below,
% Cell migration = [(Width at 0 h − Width at 24 h)/Width at 0 h)] × 100

### 2.16. Transwell Cell Migration Assay

The transwell cell migration assay chamber (24-well) containing polycarbonate membrane inserts (of 8 μm pore sizes) was used to perform the assay (Corning, Inc., Corning, NY, USA). Prior to the seeding, MDA-MB-231 cells were incubated in serum-free media with or without compounds for 6 h. Cells were seeded into the inserts and 20% serum-supplemented media was added to the bottom chamber. At the 24 h timepoint, the migratory cells were stained and counted. The data presented are an average of triplicates and two independent experiments were performed.

### 2.17. Transwell Cell Invasion Assay 

The invasion assay was performed in a transwell invasion chamber (24-well) with Matrigel-coated inserts. MDA-MB-231 cells were treated with the compounds for 6 h, trypsinized, and seeded onto the invasion chamber at a density of 20,000 cells/well. The bottom chamber was filled with 20% serum-supplemented media. The invaded cells were visualized upon staining at the 24 h timepoint and counted. The mean of the triplicate values was presented for two independent experiments. 

### 2.18. Angiogenic Tube Formation Assay

MDA-MB-231 cells were treated with the test compounds for 24 h and the conditioned media was collected. E9 (Eahy926) endothelial cells (second passage) were grown on a 6-well plate. Prior to the assay, MDA-MB-231 cells were serum-starved for 18 h, trypsinized, and seeded at a density of 10,000 cells/well onto a Matrigel-coated 96-well plate in duplicates. E9 cells with VEGF (10 ng/mL of media) were considered positive controls for the study. Tube formation was monitored at 2 h intervals and the respective images were captured. 

### 2.19. Animals and Treatment Groups

Mice were sheltered in the animal facility of Sri Ramachandra Medical College and Research Institute (Chennai, India), which has been certified as a pathogen-free animal house for performing mice experiments. All the methods were carried out according to the IAEC guidelines and regulations. Experimental procedures were framed and approved by the Institutional Animal Ethical Committee (IAEC) (IAEC approval number: IAEC/57/SRIHER/639/2019). The entire study was in accordance with the ARRIVE guidelines to ensure the safety of the animals. The sizes of the mice groups were chosen in accordance with statistical significance; the measurements and analysis were performed in an unbiased manner.

4–6 week-old female athymic mice were transplanted with MDA-MB-231 triple-negative breast tumor cells. The growth media with MDA-MB-231 cells and Matrigel (BD Biosciences) at a 1:1 ratio was mixed and injected into mice. The tumor lengths and widths were measured with an electronic caliper 2–3 times a week. The tumor volume was calculated using the formula: (length × width^2^)/2. All of the mice were randomized after 10 days, as the tumor volume reached approximately 80–90 mm^3^. A total of 25 mice were sorted into 5 treatment groups, followed by a compound treatment (SM216, SM396, SM949, and ABT199) or vehicle (*n* = 5 for each group). The drug compounds were formulated in PBS and administrated as *i.p* doses (100 mg/Kg w) for two schedules per week. Tumor growth inhibition (TGI_max_) was calculated using the following equation: (1)$$\mathrm{Tumor}\;\mathrm{Growth}\;\mathrm{Inhibition}\;=1-\frac{\mathrm{mean}\;\mathrm{of}\;\mathrm{volume}\;\mathrm{of}\;\mathrm{treated}\;\mathrm{at}\;\mathrm{day}\;x-\mathrm{mean}\;\mathrm{of}\;\mathrm{volume}\;\mathrm{treated}\;\mathrm{at}\;\mathrm{day}\;0}{\mathrm{mean}\;\mathrm{of}\;\mathrm{volume}\;\mathrm{of}\;\mathrm{control}\;\mathrm{at}\;\mathrm{day}\;x-\mathrm{mean}\;\mathrm{of}\;\mathrm{volume}\;\mathrm{of}\;\mathrm{control}\;\mathrm{at}\;\mathrm{day}\;0}\times 100$$

As the tumor size exceeded 600 mm^3^, we euthanized the mice for ethical reasons. The euthanasia procedure used was carbon dioxide asphyxiation followed by cervical dislocation. The mean of the tumor volume ± SEM for different time points was represented until we sacrificed at least one mouse in a group.

### 2.20. Immunohistochemistry of Treatment Groups—Animal Tissues 

The 5 μm-thick formalin-fixed paraffin-embedded (FFPE) tissue sections of athymic mice from different treatment groups (Vehicle, SM396, ABT199) were obtained from the Department of Pathology, SRIHER. Immunohistochemistry for proliferation marker using pre-diluted Anti-PCNA (PathnSitu Biotechnologies, Telangana, India) produced in a mouse, and the Anti-CASP3 (CSB-PA05689A0RB, CusaBio Technologies LLC, Houston, TX, USA) produced in a rabbit was performed. A standard laboratory protocol was followed. The slides were dewaxed in graded xylenes and consecutively hydrated in graded alcohols. Antigen retrieval with heated TRIS EDTA buffer (pH 9.0) was carried out, after which, hydrogen peroxide blocking, super sensitive buffer wash (Bio-genex LaboratoriesFremont, CA, USA), and protein blocking (Diagnostic BioSystems, Pleasanton, CA, USA) was conducted. The slides were labeled overnight in a moist chamber with primary antibodies. The next day following the buffer washes, the sections were incubated with the HRP polymer universal kit for secondary antibodies (Diagnostic BioSystems), and the reaction color was developed using a DAB substrate. The slides were rinsed in running water, counter-stained with hematoxylin, and mounted using DPX Mountant. The air-dried sections were observed under a Leica DM 2000 LED Light Microscope and IHC images were captured using the microscope software platform Leica Application Suite. The immune-stained sections were quantified by a senior pathologist using a Q-Scoring System. The Q-Score System represents a quantitative grading of the number of positive cells (P) and the intensity of staining (I) (Q = P × I). 

### 2.21. Plasma Stability and Hemolytic Assays

The human blood sample (5 mL) was subjected to centrifugation at 2000 rpm at 4 °C for 20 min, and plasma was collected. The 0.05 M PBS (pH 7.4)-diluted plasma (80%) was heat-inactivated at 56 °C for 30 min in a water bath before the compounds were incubated. Initially, 10 µM of test compounds were added to 1 mL of pre-heated plasma samples. To deproteinize 50 µL of plasma samples, 200 µL of acetonitrile was added and incubated for 90 min. The samples at incubation time points 0, 15, 30, 45, 60, and 90 min were vortexed (1 min) and centrifuged at 14,000 rpm, 4 °C for 15 min. The supernatants were analyzed in HPLC (Shimadzu LC-20AD HPLC system) with a UV/VIS detector (SDP-20AV), degasser (DGU-20A), and an autosampler (SIL-HT_A_). Acetonitrile–water with 0.1% acetic acid (40:60, *v*/*v*) was considered a mobile phase, and compounds were detected at 260 nm. The injection volume was 20 µL of the sample and the flow rate was 1 mL/min. The standard curves were plotted at peak areas vs. concentration (in a range of 0.01–1000 µg/mL) of the stock solution of the compounds.

Additionally, a hemolytic assay was performed on the human blood sample by separating erythrocytes upon centrifuging at 1500× *g* for 5 min and washing thrice with a sterile saline solution. Then, the cells were diluted to 10% in 100 mM of phosphate buffer (pH 7.4). To the erythrocyte suspension (200 μL), compounds were added based on the IC_50_ values. The entire contents were adjusted to a final volume of 1 mL with phosphate buffer and incubated for time points ranging from 0 to 12 h at 37 °C. The samples were centrifuged at 1500× *g* for 5 min and supernatants were analyzed for absorbance at 540 nm using the phosphate buffer as a blank. Moreover, 1% SDS was used as the positive control.

### 2.22. Statistical Analysis

All the data were obtained from two or three independent experiments representing the mean and SEM (standard error of the mean) for the data points. GraphPad Prism 6 was used for the statistical analysis (GraphPad Software, Inc., San Diego, CA, USA). We used an unpaired *t*-test or one-way or two-way ANOVA for identifying the statistical significance between two or more groups. The in vivo experimental data were analyzed using post hoc tests. The *p*-values, *p* ≤ 0.05, *p* ≤ 0.01, *p* ≤ 0.001, and *p* < 0.0001, considered significant, are represented as *, **, ***, and ****, respectively.

## 3. Results

### 3.1. SM216, SM396, and SM949 Are the Top Three Lead Molecules from Molecular Remodeling and Binding Analysis 

Our previous analysis of the BCL-2 chemical space (1650 compounds) revealed that rotatable bonds, hydrogen bond acceptors, and topological features are key factors to stabilize the interactions [28]. The functional groups viz. aromatic rings, tertiary amine, carboxyl, and hydroxyl groups are essential for selective binding to the BCL-2 (Figure 1a). Given this, we tailored the scaffold in such a way that it only had the active moieties maintaining the core integrity. This resulted in a pool of 8270 lead molecules from all 7 scaffolds, which, upon filtering for similarity, the pharmacophore and PAINS filters yielded 29 novel scaffolds.

Due to the limited synthetic viability, we searched the existing libraries for compounds based on the similarity of MACCS (molecular access) fingerprints. We found 29 of such compounds in an existing library (TIMTEC LLC) that satisfied all the criteria and were 95% similar to the designed scaffolds. We predicted the activity of these 29 molecules and all of the lead compounds showed low micromolar to nanomolar activity (Table 1), which allowed evaluating the authentic binding pattern. We found covalent warheads, such as tetrazole, aryl halides, and aryl fluorosulfonates in the designed library. To computationally validate the covalent binding, we performed covalent docking with the BCL-2 protein and observed a strong interaction pattern and affinity towards the BH4 domain of BCL-2. The structures (Figure 1b) and interaction patterns of the top three lead compounds are presented in Figure 1c.

We chose the top three compounds (SM216, SM396, and SM949) with high affinity and low predicted IC_50_ (nM) for further binding analysis using SPR experiments on a human BCL-2 immobilized sensor chip. The three compounds were mobilized onto the sensor chip from a concentration range of 6.25 to 100 nM. The direct binding SPR sensorgrams (Figure 1d) were fitted to determine the affinities (K_D_) (Table 2). Among them, SM216 exhibited the lowest K_D_ of 3.92 nM followed by SM396 and SM949 with K_D_ values of 6.37 nM, and 78 nM, respectively. ABT-199, a known selective BCL-2 inhibitor [22], showed an approximately 40-fold lower affinity for BCL-2 (Table 2). The orientations of the moieties in all three compounds favored the better occupancies reflecting the selectivity. Further, we validated the anti-cancer and anti-BCL-2 effects of all 29 compounds from a focused library via *in vitro* and *in vivo* assays and identified drug-like BCL-2-selective inhibitors.

### 3.2. SM396 Binds Covalently to the BH4 Domain of BCL-2

In the binding assays, we found that compound SM396 showed a strong association in terms of the response even after repetitively washing steps in the SPR analysis, which directed us to perform the time scale analysis with pre-incubation of SM396 on the immobilized BCL-2 chip. At different time points 0, 3 h, and 6 h, we found consistent responses and associations between BCL-2 and SM396 confirming the covalent association (Figure 2a). For compounds SM216 and SM949, we could not detect the covalent bond formation for the time scales considered. 

Similarly, we incubated the recombinant BCL-2 protein with SM396 for 12 h and the complex was analyzed by electrospray ionization mass spectrometry (ESI-MS) (Figure 2b). The deconvoluted mass analysis of MS/MS spectra showed an increase in the mass by 896 Da, confirming the formation of the covalent adduct. We further analyzed the site of mass addition in the protein and identified the BH4 domain (Figure 2c) as the binding site, consistent with the predicted model. Our results from the binding analysis by SPR and ESI-MS/MS confirmed that covalent modification by SM396 with enhanced affinity promoted irreversible modification in a time-dependent manner. 

Further, we validated the anti-cancer and anti-BCL-2 effects of SM396, SM216, and SM949 along with the initial library (26 compounds) by various *in vitro* and *in vivo* assays to identify the mechanisms of the identified BH4 selective inhibitors.

### 3.3. BCL-2 Profiling Shows Heterogenous Expressions in Different Cancer Cell Lines 

To discern the expression of BCL-2 in various cancer types, we examined the mRNA expression levels of BCL-2 from a large dataset of 30 cancer types pertaining to 1303 cell lines collected from the Cancer Cell Line Encyclopedia (CCLE) database [37]. Notably, we observed prominent expressions of BCL-2 in lymphoma, myeloma, leukemia, lung, cervical, and breast cancer cell lines (Appendix A). Further, we analyzed protein and mRNA expressions of BCL-2 by western blotting and qPCR for nine cancer cell lines belonging to lung (A549, NCI-H460, and NCI-H1299), cervical (HeLa), colon (HCT-116), and breast cancers (BT474, ZR-75, MCF7, and MDA-MB-231). The distribution of the BCL-2 protein and mRNA expression levels are presented in Appendix A. We found that among all nine cell lines, breast cancer cell lines showed high expressions of BCL-2 in both assays. Interestingly, MDA-MB-231, an aggressive triple-negative subtype of breast cancer, showed competent levels of the BCL-2 protein. We selected MDA-MB-231 for evaluating the designed lead compounds using various *in vitro* assays since this breast cancer subtype critically need targeted therapy. 

### 3.4. SM216, SM396, and SM949 Explicitly Kill Cancer Cells, and Not the Normal Cells 

The top 29 lead molecules, guided by the *in silico* results, were subjected to further *in vitro* validation by assessing the cellular cytotoxicity in both cancer (Figure 3a) and normal cell lines (Appendix A) using the MTT assay. Moreover, we performed MTT (Appendix A) and Annexin V assays on an additional cell line, BT549, a triple-negative breast cancer model, to validate the activity of the compounds (Appendix A). Based on the selective cytotoxicity on cancer and normal cells, we categorized the compounds into four groups G1, G2, G3, and G4, respectively, denoting highly active and non-toxic compounds, active and less toxic compounds, active and moderately toxic, and less active and toxic compounds (Appendix A). SM216, SM396, and SM949 of the G1 class showed extremely minimal effects on normal cell viability through the selectivity assessment by the MTT assay on different primary cells, namely human pancreatic ductal epithelial (HPDE) cells, human skin keratinocytes (HaCaT), human oral keratinocytes (HOK), and human mammary epithelial cells (HMEC) (Figure 3b). Therefore, the compounds were effectively cytotoxic to cancer cells with no effects on the normal cells, demonstrating anti-cancer potency. We also provided the AUC plots for the top three compounds for a better understanding of the activity (Appendix A).

Moreover, as per the NCI guidelines for compound screening on cancer cells, the top three compounds were subjected to cytotoxicity estimations on different cancer cell lines viz. lung cancer (NCI-H1299 and NCI-H460), brain cancer (BMG-1 and U-373MG), and breast cancer (MDA-MB-231 and MCF7). Among the three lead molecules, SM216 exhibited a potent killing effect on brain, lung, and breast tumor cells followed by SM949 with nanomolar activity on breast and lung cancer cells, and SM396 with nanomolar activity restricted to MCF7 breast cells alone (Figure 3c). An increased cytotoxic effect of the inhibitor, when compared to ABT-199, was observed in BT549 cells. The IC_50_ values calculated from non-linear regression of dose-response curves are 3.146667 μM/mL, 3.403 μM/mL, 2.7385 μM/mL, and 7.195733 μM/mL for SM216, SM949, SM396, and ABT199, respectively (Appendix A).

Further, we evaluated the effects of the compounds on cell proliferation, employing a BrdU cell proliferative assay on MDA-MB-231 cells, and found a significant decline in the S-phase fraction (Figure 3d). Moreover, we tested the induction of apoptosis by the compounds in MDA-MB-231 and BT549 cells and found an increase in the number of Annexin V-positive cells upon treatment when compared to the untreated cells (Figure 3e,f). Additional data depicting the effects of the top ten compounds on inducing apoptosis and cell proliferation by the Annexin V apoptosis assay and BrdU are presented in Appendix A. The representative images of the Annexin V apoptotic assay performed on BT549 cells treated with all three compounds are presented in Appendix A. The number of cells stained with Annexin V and PI in the double-positive population is presented in Appendix A. The results displayed a high Annexin V-positive and low PI staining indicating the onset of early apoptosis in cells treated with SM396 followed by SM949 and SM216. The representative flow plots and statistics of %S, G2/M, and G0/G1 cell populations of SM216, SM396, and SM949-treated MDA-MB-231 cells are presented in Appendix A (BrdU assay).

### 3.5. SM216, SM396, and SM949 Induce Membrane Depolarization and Cytochrome C Release

To investigate the mechanism of inducing apoptosis, we performed a series of experiments, namely, dynamic BH3 profiling and Cytochrome C release by ELISA in MDA-MB-231 cells treated with lead compounds. The dynamic BH3 profiling (DBP) technique provides a better understanding of sensitivity to apoptosis in terms of membrane depolarization and priming to death. We treated the MDA-MB-231 cells with SM216, SM396, and SM949, and after 12 h of treatment, JC-1 fluorescent-based DBP was performed according to the protocol previously described [36]. Based on the IC_50_ calculated from the viability assays, we determined the drug concentrations for treatment. We estimated the loss of fluorescence upon treatment with the compounds and FCCP (a known depolarizer) and calculated the percentage of depolarization for each compound; the results are presented in Figure 4a. We observed an upsurge in the percentage of depolarization with the treatment of SM949 accompanied by SM216 and SM396, indicating that the compounds induce apoptosis via membrane depolarization.

Subsequently, Cytochrome C release was monitored thoroughly by performing Cytochrome C-ELISA on the cytosolic fractions of the treated MDA-MB-231 cells. We observed an increase in Cytochrome C levels compared to the control in all of the treated samples (Figure 4b). We found a significant correlation between the depolarization profile and Cytochrome C protein levels, which substantiated the BCL-2-mediated apoptosis. Moreover, we quantified the Cytochrome C release on MCF7 cells and identified a similar trend in the concentration of Cytochrome C released after treating the cells with lead molecules (Appendix A). The results suggest that SM216, SM396, and SM949 could contribute to membrane depolarization and increased Cytochrome C levels (in cytosol).

### 3.6. SM216, SM396, and SM949 Attenuate the Transforming Capabilities of Breast Cancer Cells 

To determine the effect on cellular transformation, we carried out colony formation, angiogenesis, wound healing, anchorage independency, invasion, and migration assays on MDA-MB-231 cells treated with the compounds. The compounds suppressed the clonogenic activity of the MDA-MB-231 cells significantly by 70%, indicating cell death as a consequence of cytotoxicity (Figure 5a). 

In the soft agar assay, we assessed the cell proliferation of MDA-MB-231 cells in an anchorage-independent manner and found that the colony formation ability was reduced upon treatment with the SM396 followed by SM216 and SM949 pertaining to the control. The representative images are presented in Figure 5b. We performed transwell migration and invasion assays to analyze the impairment of the chemotactic response of cancer cells and to test the ability of cancer cells to adhere to ECM, respectively. Cell migration and invasion were attenuated in the treated cells when compared to the control (Figure 5c,d).

To determine the ability of MDA-MB-231 cells to migrate as a result of a mechanical scratch wound in the presence of SM216, SM396, and SM949, we carried out wound healing assays at different time points (0 and 24 h). The healing of the wound was monitored and the same is illustrated in Figure 5e. The scratch was completely closed within a 24 h time point in the control, whereas in the treated sample (SM396), we detected only a 35% closure, implicating that the compounds presented inhibitory effects on cell migration. In BT549 cells, the scratch was almost completely closed within the 24 h time point in the control. The treated samples, specifically SM396, only showed a 27.5% closure, implicating that the compound presented an inhibitory effect on cell migration (Appendix A).

Further, we showed that the tube-forming angiogenic ability of endothelial cells (EaHy926 cells) was radically diminished with the treatment of compounds by 70–80% compared to the control (Figure 5f). Hence, these compounds are capable of inhibiting tumor angiogenesis, which is also a BCL-2-mediated process [38]. Additionally, the results of clonogenic, soft agar, migration, and wound healing assays performed on MDA-MB-231 cells for the top 10 lead compounds are presented in Appendix A–h. Furthermore, we performed the wound healing assay for the top 10 compounds on MCF7 cells and the results are presented in Appendix A.

### 3.7. SM216, SM396, and SM949 Are Highly Tolerated by Erythrocytes at Efficacious Doses

We evaluated the compound tolerance in human blood pertaining to its stability and hemolysis by the plasma stability and hemolytic assays, respectively. We tested the stability of compounds in human plasma samples by incubating the compounds with the plasma for 0–120 min. We detected that ~70–80% of the compound was stable and active from the HPLC chromatograms (Figure 6a) of human plasma samples until 120 min of incubation.

Similarly, the blood samples were subjected to compound treatments according to IC_50_ and were incubated at different time points from 0 to 12 h (Figure 6b). We observed <5% hemolysis until 120 min of incubation and only minimal hemolysis until the 12 h time point, which demonstrated that the compounds were well tolerated and less toxic. These results directed us to perform *in vivo* studies since the compounds were stable in plasma and non-hemolytic. 

### 3.8. SM216, SM396, and SM949 Showed Anti-Tumor Activity on Human Breast Cancer Xenografts on Athymic Mice

To test the *in vivo* anti-cancer efficacy of the novel small molecules viz. SM216, SM396, and SM949, 4–6 weeks old female athymic mice were xenografted with MDA-MB-231 cells on the mammary fat pads and palpable tumors were induced. After 10 days, 25 mice bearing breast tumors were randomized into 5 groups (*n* = 5) and the compounds (SM216, SM396, and SM949) were administered as *i.p* doses at concentrations of 100 mg per kg of body weight, along with a known FDA-approved drug, ABT-199 (100 mg per kg) and the vehicle control, respectively. Tumor volumes were measured at regular intervals and the data were tabulated. The data confirmed that the novel small molecules, SM216, SM396, and SM949, inhibited tumor growth significantly compared to ABT-199. The growth kinetics and maximal tumor growth inhibition (TGI_max_) are depicted as tumor volumes (five groups of mice treated with different drugs) in Figure 6e,f. The average tumor volume for all groups on day 0 of treatment was 92.77 mm^3^. We observed that all three lead molecules inhibited tumor growth efficiently compared to the vehicle control. On day 15, upon 4 doses, the average tumor volume of the vehicle control group was measured as 470.33 mm^3^, whereas for the treatment groups, SM216, SM396, SM949, and ABT-199 were 308.59 mm^3^ (*p* ≤ 0.0001), 294.6 mm^3^ (*p* ≤ 0.0001), 334.58 mm^3^ (*p* ≤ 0.0001), and 371.94 mm^3^ (*p* ≤ 0.0001), respectively (Table 3). Further analysis comparing the tumor volumes on day 15 treated with leads SM216, SM396, and SM949 showed that all three compounds exhibited similar anti-cancer efficacy. These compounds achieved regression of 60% after 20 days of treatment. Notably, among the three compounds, SM396 efficaciously inhibited tumor growth, evident in terms of its TGI_max_ (57%). Moreover, these compounds were well tolerated with no weight loss and hemolysis in all treatment groups. The survival was 100% for all treatment mice groups until the termination of the study. The breast tumor cells of athymic mice from different treatment groups demonstrated nuclear expressions of proliferation marker (PCNA). A decrease in the expression of PCNA was observed upon treatment with SM396 compared to the vehicle control. Increased cytoplasmic staining of Caspase 3 was observed in the SM396-treated group, indicating the activation of apoptotic machinery (Figure 6c,d). The mean q score was calculated for SM396 and ABT-199 IHC samples and the results are tabulated in Appendix A. SM396 and ABT-199 showed a similar intensity (3+) and a mean q score. 

## 4. Discussion

TNBC cases are mostly insensitive to therapeutic regimens, including anthracyclines, alkylating agents, and taxanes. There is a desperate need for improved therapeutic modality in this subset of cancer and targeted therapies may address this clinical conundrum. Several studies have portrayed BCL-2 as a significant prognostic marker in luminal A, ER-positive subtypes of breast cancer [39,40]. Various small molecules capable of mimicking the BH3-binding sites and absolving the pro-apoptotic proteins were reported [41,42,43,44,45,46]. Among them, ABT-199, ABT-737, etc., are reported to sensitize tumor cells towards drugs, such as cisplatin, doxorubicin, and paclitaxel [47], but failed as single agents exhibiting resistance [30]. These findings provide a strong rationale for our study in designing novel scaffolds with high affinity and the BH4 domain as an ideal target to circumvent the resistance.

Our analysis dealt with a double-edged sword with the BH4 domain as a novel site for the targeted inhibition of BCL-2, on one side, and triple-negative breast cancer on the other side, and proposed covalent modification as a plausible solution for tackling these two challenges. The proposed strategy provided potentiating evidence that the compounds are selective BH4 mimetics in biophysical, *in vitro* and *in vivo* assays. We hypothesized that reshuffling the scaffolds (apogossypol, thiomorpholine pyrazole, benzothiazole, quinolone, quinazoline, and polyquinoline) would induce strong binding to BCL-2. The SPR data are consistent with the predicted data claiming the top three compounds as the best hits. Moreover, remodeling by the addition of functional groups, such as tetrazoles and trifluorides [29], enhances the pharmacological efficacy and reduce the toxicity of the compounds. This was evident from the results of plasma stability and hemolytic assays performed on human blood samples. The affinity studies using the SPR system revealed covalent binding (nanomolar affinity) of SM396 and reversible binding of SM216 and SM949 towards BCL-2. Our study is a novel attempt in designing potent covalent and non-covalent lead molecules showing nanomolar activity in binding to the BH4 domain of BCL-2. 

The data from the follow-up experimental studies strongly suggest that the compounds SM216, SM396, and SM949, are effective in inducing mitochondrial-mediated apoptosis specific to cancer cells with no killing effect on normal cells. The compounds can induce apoptosis by membrane depolarization and increased Cyt C levels in the cytosol. We plan to further explore the detailed mechanism of action as part of future directions to the work. Moreover, the compounds are efficacious in inhibiting the prominent characteristics of cancer cells. Altogether, targeting TNBC cells with the compounds would achieve killing and inhibitory effects on tumor progression and metastasis. Furthermore, the compounds exhibited cytotoxicity in lung and glioma cancer types. We speculate that BCL-2 intervention with these agents in TNBC, along with solid tumors viz. the lung and brain, might prove beneficial.

Further, our preclinical studies presented tumor growth inhibition in athymic mice presumably due to selective inhibition of BCL-2. The data suggest that BH4 mimetic therapy appeared to be well tolerated, as body weight and hematopoietic parameters were maintained, establishing the pharmacological competence of the compounds. Despite the high binding affinity of the compounds towards the BCL-2 protein, the *in vivo* activities of the compounds which are dependent on both cellular and pharmacological factors, would be notable constraints in achieving high activity compared to ABT-199. In this study, we found that three novel compounds exhibiting anti-BCL-2 activity as single agents could be promising agents for BCL-2-driven personalized therapy in TNBC patients.

## 5. Conclusions

Here, we proposed three highly potent and novel BH4 mimetics (covalent and non-covalent) identified by a robust *in silico* remodeling of the BCL-2 chemical space. The data demonstrated remarkable single-agent anti-tumor activity in all *in vitro* and *in vivo* investigations, sparing the normal cells at a nanomolar dosage. We confirmed that the compounds potently induced cell death through apoptosis in TNBC, which is challenging to target compared to ABT-199. To circumvent the pharmacological drawbacks, our study also presented the first-in-class covalent BH4 mimetics, which gratified the need of the pharmacological arena in targeting challenging cancer types.

## Figures and Tables

**Figure 1 cancers-14-05241-f001:**
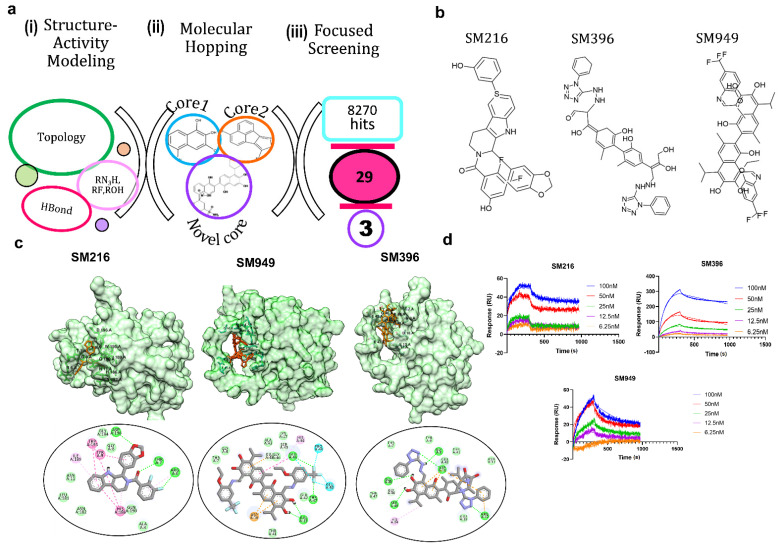
**Molecular remodeling and binding of novel scaffolds to BCL**−**2**. (**a**) In silico methods for identifying potential BCL−2 inhibitors: (i) structure–activity modeling to relate the activity of BCL-2 inhibitors with physicochemical descriptors, such as topological parameters, hydrogen bonding descriptors, and functional groups, (ii) molecular hopping for designing novel scaffolds by combining two cores from seven distinct core libraries, and (iii) focused screening of 8270 compounds generated from molecular hopping, which resulted in a set of 29 compounds and narrowed down to three. (**b**) Structures of the top three lead compounds SM216, SM396, and SM949. (**c**) Structures of BCL−2 bound to SM216, SM396, and SM949 at the BH4 domain, respectively, and interactions are shown below. (**d**) SPR sensorgrams depicting the binding kinetics of the compounds to the BCL−2 protein are shown. (Drug concentrations used range from 6.25–100 nM).

**Figure 2 cancers-14-05241-f002:**
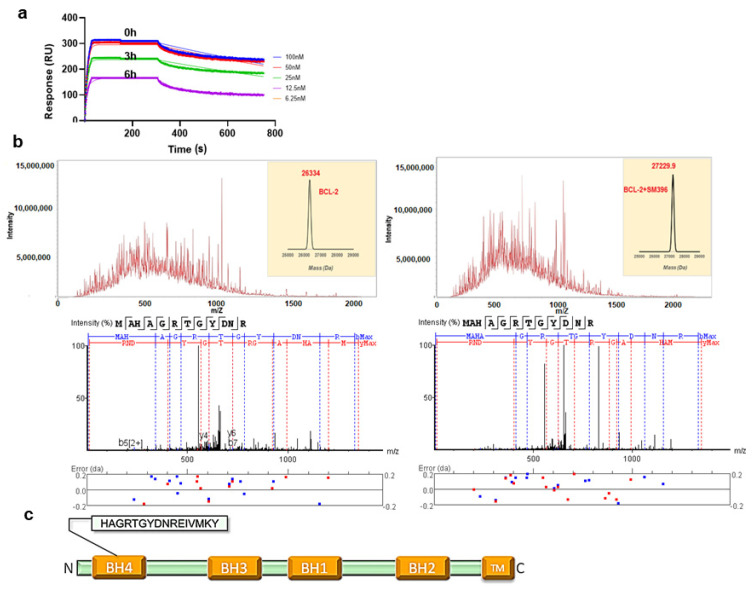
**SM396, a covalent BH4 inhibitor.** (**a**) The SPR sensorgrams for the interaction between immobilized BCL-2 and SM396 upon incubation with 1 μM of SM396 for the indicated times of 0 h, 3 h, and 6 h. (**b**) Raw spectra and deconvoluted mass weights of molecular weights (right) for unmodified and SM396-modified BCL-2 are shown. MW (unmodified, calc): 26,334; MW (modified, calc): 27,229.7 Da with mass addition of 895.7 Da corresponding to SM396. Representative MS2 spectrum corresponding to BCL-2 is shown below, with b+ and y+ peaks annotated. The observed peaks are consistent with the BH4 domain sequence of human BCL-2. (**c**) Domain structure of BCL-2 with BH4 domain sequence shown.

**Figure 3 cancers-14-05241-f003:**
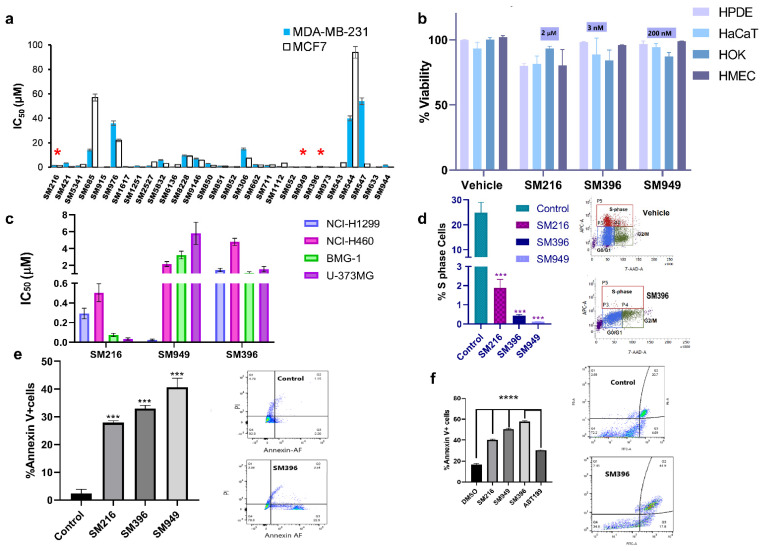
**BH4 mimetics SM216, SM396, and SM949 are effective in inducing apoptosis in breast cancer cell lines.** (**a**) The IC_50_ values estimated from the dose-dependent inhibition curves of 29 lead compounds using the MTT assay (72 h timepoint) for MDA-MB-231 and MCF7 breast cell lines (SM216: 1.866 μM, SM396: 2.995 nM, and SM949: 0.1875 μM), (*n* = 3, *p <* 0.001). (**b**) The percentage of cell viability values for normal cells (HPDE, HaCaT, HOK, and HMEC) treated with SM216, SM396, and SM949. The data are represented as mean viability ± SEM (*n* = 3, *p* < 0.05). (**c**) The IC_50_ values estimated from the dose-dependent inhibition curves of three lead compounds using an MTT assay (72 h timepoint) for lung cancer cells (NCI-H1299 and NCI-H460) and glioma cells (BMG-1 and U-373MG). (**d**) BrdU cell proliferation assay on MDA-MB-231 cells showing a reduction in % S-phase fraction for SM216, SM396, and SM949, and the vehicle control and representative dot plot for SM216 are shown. (**e**) The percentage of Annexin V + positive cells for SM216, SM396, and SM949 along with representative images from FACS analysis on MDA-MB-231 cells. The data are represented as mean ± SEM for all the assays. (**f**) Bar graph showing the percentage of apoptotic cells upon treatment with the vehicle control, SM216, SM396, SM949. An increase in the apoptotic cell fraction upon SM396 treatment is observed by Alexa Flour 488 Annexin V-PI assay using flow cytometry on BT549 cells at 6 h time-points shown in the representative pictures. *, *** and **** indicate *p* ≤ 0.05, *p* ≤ 0.001 and *p* ≤ 0.0001, respectively.

**Figure 4 cancers-14-05241-f004:**
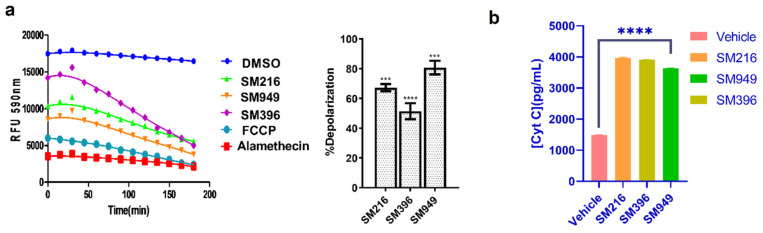
SM216, SM396, and SM949 induce apoptosis via mitochondrial membrane depolarization and Cytochrome C release in MDA-MB-231 cells. (**a**) Dynamic BH3 profiling assay using JC-1 staining on MDA-MB-231 cells treated with the compounds. (**b**) Analysis of Cytochrome C release (Cyt C) by ELISA. The dosage used for the assay was as follows, SM216: 1.866 μM, SM396: 2.995 nM, and SM949: 0.1875 μM. ****, *** represent *p* ≤ 0.0001 and *p* ≤ 0.001, respectively.

**Figure 5 cancers-14-05241-f005:**
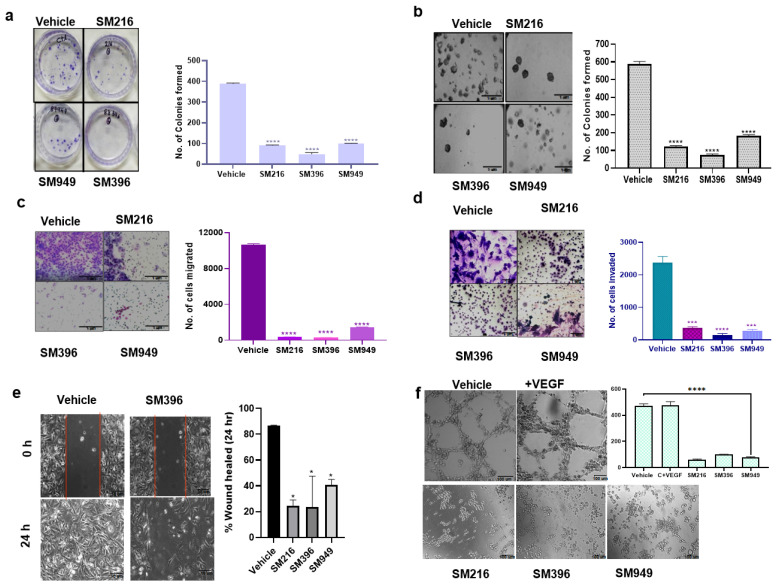
**SM216, SM396, and SM949 effectively inhibited the tumorigenic potential of MDA-MB-231 cells.** (**a**) Clonogenic assay showing the reduced number of colonies formed in MDA-MB-231 cells upon treatment (*n* = 2). (**b**) Soft agar assay showing reduced colony formation of MDA-MB-231 cells upon treatment. (**c**,**d**) Transwell assay showing reduced migration and invasion capabilities of MDA-MB-231 cells upon treatment. (**e**) Wound healing assay showing reduced migration of MDA-MB-231 cells upon treatment. (**f**) Angiogenesis assay showing reduced tube formation of EaHy976 cells upon treatment. (Drug concentrations used SM216: 1.866 μM, SM396: 2.995 nM, and SM949: 0.1875 μM). * denotes *p* ≤ 0.05, *** denotes *p* ≤ 0.001 and **** denotes *p* ≤ 0.0001.

**Figure 6 cancers-14-05241-f006:**
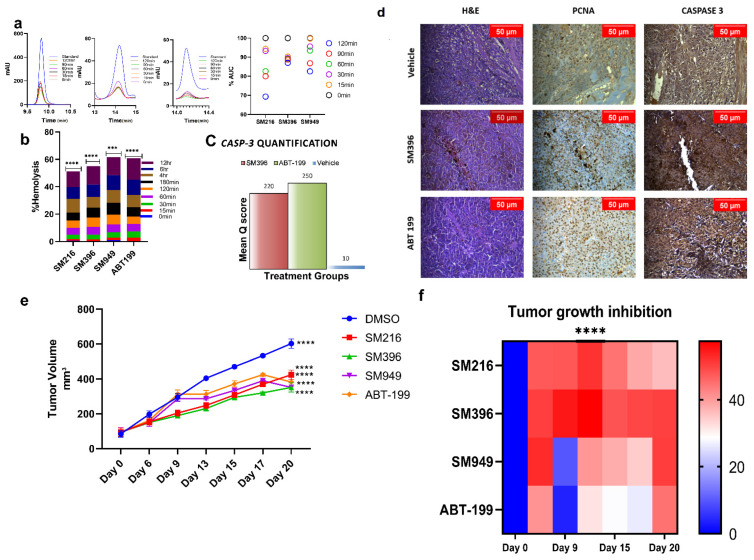
**In vivo anti-tumor activity of SM216, SM396, and SM949 on MDA-MB-231 xenografts.** (**a**) HPLC chromatograms of the compounds incubated in human plasma at different time points (0–120 min). (**b**) Bar graph showing the percentage of hemolysis (using hemolytic assay) at 0–12 h upon treatment. (**c**) Caspase-3 quantification based on mean q score (percentage of positive stained cells × Intensity of the stain) in animal tissue samples exposed to different treatments. (**d**) Photomicrographs showing the expressions of proliferation markers—PCNA and Caspase 3 staining in athymic mice mammary tissues upon treatment with vehicle control, small molecule inhibitor SM396, and positive control ABT-199 at 20X magnification. A decrease in the expression of proliferation markers and an increase in Caspase 3 expression was observed upon treatment with SM396. (**e**) Growth kinetics depicting the anti-tumor activities of the compounds and representative images of excised tumor xenografts at the end of the study. (**f**) Heat map showing the tumor growth inhibition (TGI_max_). The stars indicate the *p*-value significance of the two-way ANOVA performed with control (ABT-199) samples. (Drug concentrations used SM216: 1.866 μM, SM396: 2.995 nM, and SM949: 0.1875 μM). *** = *p* ≤ 0.001, **** = *p* ≤ 0.0001.

**Table 1 cancers-14-05241-t001:** Experimental, predicted activity, and binding energy of the compounds. The data for top three compounds was shown in bold and highlighted in light red color.

Lead Compounds	Predicted IC_50_ (µM)	Experimental IC_50_ (µM)	Binding Energy (kcal mol^−1^)
**SM216**	**1.2**	**1.602**	**−9**
SM421	2.9	3.38	−5
SM5341	1.4	1.183	−4.4
SM685	13	14	−5.2
SM915	0.00039	0.0004	−4.7
SM976	32	36	−5.8
SM1617	0.65	0.74	−3.7
SM1251	1.6	1.15	−4.9
SM2527	2.1	1.16	−4.6
SM5832	6.1	5.923	−3.4
SM6136	13	0.178	−3.9
SM8228	41	9.49	−4.9
SM9146	10	7.09	−3.2
SM850	4	3.002	−4.5
SM851	1.7	0.88	−5.6
SM852	0.47	0.423	−5.9
SM306	6	14.9	−6.5
SM662	2.1	2.054	−6.8
SM711	1.8	1.58	−4.3
SM1112	0.567	0.654	−4
SM652	0.324	0.373	−5.9
**SM949**	**0.18**	**0.172**	**−8.6**
**SM396**	**0.0016**	**0.00273**	**−9.3**
SM973	0.23	0.201	−8.9
SM543	0.0016	0.00182	−7.6
SM544	31	40	−8.3
SM547	56	54	−5.7
SM633	0.013	0.006	−4.6
SM944	1.8	1.795	−10.8

**Table 2 cancers-14-05241-t002:** Binding affinities of SM216, SM396, SM949, and ABT-199 for BCL-2 (surface plasmon resonance).

Compound	K_D_ (nM)
SM216	3.92
SM396	6.37
SM949	78
ABT-199	116

**Table 3 cancers-14-05241-t003:** Mean tumor volumes of athymic mice with MDAMB-231 xenografts after treatment.

Groups	Tumor Volume (mm^3^)		Tumor Volume (mm^3^)		Tumor Volume (mm^3^)		Tumor Volume (mm^3^)		Tumor Volume (mm^3^)		Tumor Volume (mm^3^)		Tumor Volume (mm^3^)	
	Day 0		Day 6		Day 9		Day 13		Day 15		Day 17		Day 20	
	Mean	SEM	Mean	SEM	SEM	SEM	Mean	SEM	Mean	SEM	Mean	SEM	Mean	SEM
DMSO	84.84	19.16	197.77	19.33	295.63	21.97	404.38	3.78	470.33	7.94	532.7	6.09	602.01	5.85
SM216	94.01	23.93	153.57	17.17	204.5	16.69	248.27	16.37	308.59	12.10	369.88	13.0	424.26	11.41
SM396	94.04	24.32	150.29	12.83	189.48	11.53	230.27	11.32	294.6	3.09	320.51	5.32	351.81	4.68
SM949	95.6	25.45	149.12	21.54	286.11	15.57	286.11	6.63	334.58	3.68	389.36	3.57	351.73	88.06
ABT-199	95.35	23.42	162.15	20.46	313.94	23.32	313.03	20.43	371.94	17.45	424.83	9.64	382.6	95.99

## Data Availability

All of the study data are available from the corresponding author.

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
