# Peer review of "Novel BH4-BCL-2 Domain Antagonists Induce BCL-2-Mediated Apoptosis in Triple-Negative Breast Cancer"

_cancers, 2022, doi:10.3390/cancers14215241_

Round 1

Author Response

Major concerns:

On page 2, line 76, the authors state that "Our previous analysis on the BCL-2 chemical space (1650 compounds) revealed that..." (26). In this reference 26, none of the authors listed in the current manuscript were involved. There was no information about the screening of 1650 compounds. Also, the manuscript was published in 2019, Mar; 9(3): 342-353. PMID: 30514704 but it is mentioned in the manuscript as 2018.

We thank the reviewer for mentioning this. We apologize for the oversight. As suggested the appropriate reference is now added. (p8, line 351).                                                                                                           

On page 4, lines 114 & 115, "ABT-199, a known selective BCL-2 inhibitor [21] showed an approximately 40-fold lower affinity for BCL-2 (Figure 1d)" but there are no results were shown in Figure 1d for the ABT-199, shown only for SM216, SM396, and SM946 at different concentrations (6.25 nM to 100 nM). However, in Table 2, they have shown a low affinity for ABT-116 (116 nM) vs 3.92 nM for SM216.                                                       

We thank the reviewer for this suggestion. The sentence has been revised (p10, 388).

On page 5, Figure 2a, the Time unit in X-axis is missing (sec, min, or h), and also no information about the different color lines. In Figure 2b, both the X and Y-axis units are not readable.

We thank the reviewer for this comment and we sincerely apologize. The Figure is now revised giving all the details about the Units/Axes. (Fig 2b)

Somehow, the supplementary Figure 1 is not readable, could not be opened.

We apologize for this in convenience, We have replaced it and also provided a Tiff image format.

In Figures 3e and 3f, there is no labeling of the cell lines. Hopefully, the data for MDA-MB-231 would be Figure 3e and the data for BT549 would be Figure 3f.

The authors thank the Reviewer for this oversight from our side, now this has been rectified by adding the cell line names. (Figure 3)

On page 8, line 207, the authors mentioned that they used ABT-199 as a positive control but no data was presented for ABT-199 in the entire panels in Figure 3.

The authors thank the Reviewer for this error from our side, we would like to humbly state that ABT-199 was not used in all the preliminary experiments presented in Figure 3. ABT 199 was used for in vivo studies and as a control on another TNBC cell line, BT 549 and the data on BT 549 cell line is presented in supplementary figure 4d.

In Figure 6, the expression of BCL-2 is missing in vehicle and compounds-treated mice tumors, either immunoblot or immunohistochemistry.

The authors thank the reviewer for this comment. At this point of time, our in vivo study was performed to give a proof of concept for the anti-tumor activity of the novel compounds, hence the end points for the study were looking at only the anti-proliferative and tumor inhibition activity. Thus, we stained for proliferation marker PCNA and apoptotic marker Caspase 3 to show the apoptosis induction.

We apologize that, we haven’t checked the BCL-2 expression in tissue sections, as the compounds are targeting the protein-protein interaction of the BCL-2 and subsequent induction of apoptosis, thereby BCL-2 level is likely to remain unaltered upon treatment.

Minor concern:

The abstract needs to be simple and understandable.

The authors thank the reviewers for suggestion and as stated the abstract has been modified.

Reviewer 2 Report

This manuscript provides a good understanding of the efficacy of BH4 domain-bound Bcl-2 inhibitor small molecule compounds in vitro in vivo experiments against triple-negative breast cancer compared to ABT-199. My comments are as follows.

1.     Small molecules inhibiting the BH4 domain showed greater growth inhibition than small molecules inhibiting the BH3 domain by inducing apoptosis. What mechanism could explain the difference in proliferation inhibition in these compounds?

2.     In xenograft model, the growth-inhibitory effect of Bcl-2 inhibitors did not lead to tumor shrinkage as a single agent, but rather to the need for combination therapy with Bcl-2 inhibitors and anticancer agents in clinical situations.

3.     Strategies targeting cancer stem cells are needed in the clinical setting in terms of breast cancer cure; are small molecules of Bcl-2 inhibitors effective against breast cancer stem cells and do they overcome drug resistance? Given these circumstances, inhibition of autophagy may be more important than inhibition of apoptosis in cancer stem cells.

4.     In clinical trials, it is important to have fewer adverse events; could small molecules targeting the Bcl-2 BH4 domain advance to clinical trials in triple-negative breast cancer?

Author Response

Small molecules inhibiting the BH4 domain showed greater growth inhibition than small molecules inhibiting the BH3 domain by inducing apoptosis. What mechanism could explain the difference in proliferation inhibition in these compounds?

                  The authors thank the reviewer for this insightful comment. We would like humbly state that literature provides evidence on importance of BH4 domain in sustaining the anti-apoptotic function of BCL-2 family proteins. However, BH3 mimetics are specific and induce apoptosis only when binding to pro-apoptotic members whereas BH4 mimetics have multiple protein partners (Ca2+ ion signalling, nuclear translocation and associated pathways) involved in inducing apoptosis. (This understanding is based on the published paper:- https://doi.org/10.1046/j.1365-2184.2003.00252.x)

  1. In xenograft model, the growth-inhibitory effect of Bcl-2 inhibitors did not lead to tumor shrinkage as a single agent, but rather to the need for combination therapy with Bcl-2 inhibitors and anticancer agents in clinical situations.

                  The authors thank the reviewer for this comment, based on literature the inhibitory effect of BCL-2 inhibitors is highly dependent on level of apoptotic priming in the tumor cells. Thus, our  in vivo experiments showed tumor shrinkage as single agents suggesting apoptotic priming, based on tumor shrinkage, Caspase 3 expression, etc. Thus we speculate that these novel compounds could also be used as combination therapy depending on their genetic context.

  1. Strategies targeting cancer stem cells are needed in the clinical setting in terms of breast cancer cure; are small molecules of Bcl-2 inhibitors effective against breast cancer stem cells and do they overcome drug resistance? Given these circumstances, inhibition of autophagy may be more important than inhibition of apoptosis in cancer stem cells.

                  The authors thank the reviewer for this comment, however the present study is a proof of concept to show the anti-tumor activity of these novel compounds. Thus we did not perform experiments using these compounds to show their effect on cancer stem cells. However, we will carry out these studies in future. Further, based on literature evidence, BH4 domain was shown to induce both apoptosis and autophagy in a context dependent manner  (https://doi.org/10.1593/neo.121392), so we hope that targeting BH4 domain is more beneficial in achieving a prominent killing effect than BH3 domain.

  1. In clinical trials, it is important to have fewer adverse events; could small molecules targeting the Bcl-2 BH4 domain advance to clinical trials in triple-negative breast cancer?

The authors thank the reviewer for this insightful comment. Our preliminary in vitro experiments using these compounds on normal cells and erythrocytes did not show any toxicity. However further detailed in vivo acute and chronic toxicity studies are needed for confirmation.

Reviewer 3 Report

Kanakaveti et al. identified three novel BH4 mimetics and characterized their binding pattern and kinetics. Furthermore, the authors validated their killing effects using TNBC models in vitro and in vivo showing these compounds could effectively induce apoptosis and inhibit TNBC cell transformation. The findings are novel and may present potential therapeutic value however some parts of the study design and technical pitfalls significantly impaired the credibility of the observations, as listed below.

*Figure 3a, please use AUC instead of EC50 and show the curve consisting of different concentrations.

*Figure 3b, what are the concentrations used in normal cells? Show the killing curve with at least 5 concentrations (covering the range used for malignant cells) and calculate AUC to compare.

*Figure 3e, compensation problem or cytometer setup problem. False claim.

*Figure S4b, in the illustration of Annexin V-PI staining, compensation is clearly off as quite a few cells showed PE positive(Q2) in Annexin single staining. PI is a strong stain in the PE channel yet the flow plots showed intensity between 10^2 to 10^3, suggesting the voltage was not set correctly. Check the flow plot out in the manufacturer's guidebook(https://www.thermofisher.com/document-connect/document-connect.html?url=https://assets.thermofisher.com/TFS-Assets%2FLSG%2Fmanuals%2Fmp13241.pdf). 

*Figure S4c, why the authors stained Annexin V and PI separately and how could it be possible to estimate double positive cells as they stated in line 218?

*Figure 3d, indicate how S phase is calculated, and show the flow plots in addition to statistics in Sup *Fig2d. What does the right flow panel indicate? How could one even overlay two flow plot side by side on the same x and y axles?

Figure 3f, vehicle control showed >50% apoptotic cells, suggesting something is significantly wrong with this experiment, eg. contamination, wrong culture condition, etc.

Line 211. Annexin V+ ve?

Line 223. Vis.,???

*Line 221-249. The authors claim the apoptosis these three agents induced is BCL-2 dependent based on their observation of cytochrome C release and membrane depolarization. Of note, these two assays could only indicate apoptosis happened. KO/KD experiments and rescues are needed to verify BCL-2 dependency. 

Figure 5a, please indicate how many plates you did. In addition, the concentration used and the rationale for using that concentration.

Line 290, human blood cells is vague and inaccurate. RBC only

Figure 6. in vivo studies, please explain why these compounds were given through i.p.

*Figure 6e bottom. a scale by the tumor is required to visualize tumor mass.

*Figure 6d, authors need to show the section area staining by H&E, PCNA, and caspase3. Otherwise, there is no way to compare them.

In the introduction, please discuss the function of BH4 domain and its binding proteins and function beyond BH3 proteins.

Author Response

Reviewer 3

Kanakaveti et al. identified three novel BH4 mimetics and characterized their binding pattern and kinetics. Furthermore, the authors validated their killing effects using TNBC models in vitro and in vivo showing these compounds could effectively induce apoptosis and inhibit TNBC cell transformation. The findings are novel and may present potential therapeutic value however some parts of the study design and technical pitfalls significantly impaired the credibility of the observations, as listed below.

We thank the reviewer for the positive and constructive comments.

*Figure 3a, please use AUC instead of EC50 and show the curve consisting of different concentrations. Figure 3b, what are the concentrations used in normal cells? Show the killing curve with at least 5 concentrations (covering the range used for malignant cells) and calculate AUC to compare.

The authors thank the reviewer for this valuable suggestion. We have now calculated the   AUC for top 3 compounds and presented as supplementary FigureS5. We would like to add that concentration range of 1nM- 10 µM was used for both cancer cells and normal cells.

*Figure 3e, compensation problem or cytometer setup problem. False claim.

The authors apologize for the ambiguity. We would like to humbly state that the experiments were carried out on BD Flow cytometer which has an auto-compensation with in default settings. Additionally, as suggested by the expert reviewer, we performed manual compensation for the fcs files and the plots are now provided in the revised Figure 3e.

*Figure S4b, in the illustration of Annexin V-PI staining, compensation is clearly off as quite a few cells showed PE positive(Q2) in Annexin single staining. PI is a strong stain in the PE channel yet the flow plots showed intensity between 10^2 to 10^3, suggesting the voltage was not set correctly. Check the flow plot out in the manufacturer's guidebook(https://www.thermofisher.com/document-connect/document connect.html?url=https://assets.thermofisher.com/TFS-Assets%2FLSG%2Fmanuals%2Fmp13241.pdf). 

The authors thank the reviewer for this comment. In this study, Annexin V staining was performed after 4 hours of treatment with the compounds. This was done to capture the cells at the early apoptotic phase and check the efficacy of the compounds in triggering Apoptosis. Further, we would like to humbly state that the experiment was carried out according to the manufacturer’s instructions. Untreated unstained and untreated single stained (only Annexin V and only PI) were used for gating and compensation. We sincerely hope this clarifies the data presented.

*Figure S4c, why the authors stained Annexin V and PI separately and how could it be possible to estimate double positive cells as they stated in line 218?

We apologize for the ambiguity in expressing the results. We would like to state that the staining wasn’t done separately but the population distribution of Annexin and PI positive cells in the sample were plotted separately. The Annexin V apoptosis assay was done according to the manufacturer’s protocol. Further, the readout for our experiment is to capture the cells undergoing early apoptosis, that is, when PI stains and also allows Annexin to enter the cells and additionally stain the cytosolic side of the plasma membrane. Therefore, Annexin PI stained (double positive) apoptotic cell populations always show a population which is high in Annexin and low in PI as presented in the figure. SM396 showed increased level of Annexin V positive cells when compared to other treated samples (highlighted in the MS).

*Figure 3d, indicate how S phase is calculated, and show the flow plots in addition to statistics in Sup *Fig2d. What does the right flow panel indicate? How could one even overlay two flow plot side by side on the same x and y axles?

We apologize for the ambiguity in representing the results. We have now revised all the information in the figure, corrected the merged axes and is presented in Figure3D

Figure 3f, vehicle control showed >50% apoptotic cells, suggesting something is significantly wrong with this experiment, eg. contamination, wrong culture condition, etc.

We agree with reviewer, this could be possible due to our vehicle control concentration (10mM) used for the assay. We anticipate this could a probable reason for the observed result. However, we present the normalized apoptotic rate from control and treated samples to show the effect of the compounds. Kindly refer to following paper for more details on apoptotic rate PMID: 12210142.

Line 211. Annexin V+ ve?

The authors apologize for this error, the line refers to BRdU assay, in which the 7AAD fraction of the samples were discussed.

Line 223. Vis.,???

As suggested by the expert reviewer, we have now modified the sentence.

*Line 221-249. The authors claim the apoptosis these three agents induced is BCL-2 dependent based on their observation of cytochrome C release and membrane depolarization. Of note, these two assays could only indicate apoptosis happened. KO/KD experiments and rescues are needed to verify BCL-2 dependency. 

The authors thank the insightful comment of the expert reviewer and agree the need for KO/KD experiments. We are planning additional experiments as future work for this project.

Figure 5a, please indicate how many plates you did. In addition, the concentration used and the rationale for using that concentration.

The authors thank the expert reviewer for this observation. In this study, we used duplicates for each concentration and the experiments were repeated twice (n=2), we choose the IC50 values calculated from the MTT assays for each compound as treatment concentration for all the assays to test the efficacy of the compounds.

Line 290, human blood cells is vague and inaccurate. RBC only

The authors apologize for this ambiguity. The sentence was rephrased as suggested. (line290)

Figure 6. in vivo studies, please explain why these compounds were given through i.p.

The authors thank the reviewer for this important note. The xenograft model is a robust model and is well suited for proof of concept study. The investigational compounds were injected through the IP route for greater drug absorption, ease of administration and importantly less trauma to the experimental animals.

*Figure 6e bottom. a scale by the tumor is required to visualize tumor mass

The authors apologize for this oversight, however we would like to state that we measured the tumors using a digital Vernier Calipers and wet weights of excised tumors using a weighing balance. We apologize for not using a scale while taking the photographs. We ensure to use it in future experiments.

*Figure 6d, authors need to show the section area staining by H&E, PCNA, and caspase3. Otherwise, there is no way to compare them.

We thank the reviewer for the comment and we have included the H&E stained photomicrographs in the panel.

In the introduction, please discuss the function of BH4 domain and its binding proteins and function beyond BH3 proteins.

We thank the reviewer for the comment we have now included the function of BH4 and its potent interactions in the introduction.

Round 2

Reviewer 1 Report

The authors are well responsive to the previous critics and the current version of the manuscript is great for the readers.

Author Response

We thank the reviewer for accepting our manuscript.

Reviewer 3 Report

Point-to-point response review

*Figure 3a, please use AUC instead of EC50 and show the curve consisting of different concentrations.

The authors thank the reviewer for this valuable suggestion. We have now calculated the   AUC for top 3 compounds and presented as supplementary FigureS5. We would like to add that concentration range of 1nM- 10 µM was used for both cancer cells and normal cells.

The authors added a dose-repone curve of top three compounds. However, some compounds are not potent to reach 50% viability (IC50) eg. MCF7+SM396 and MDA+SM216 treatment combinations. Therefore, the authors must have imputed the IC50 based on the dose curve, which is inaccurate and misleading. Also, Fig S5 needs modifications (inconsistent front&size, style, and compound name is missing). The authors stated they calculated AUC yet results are not included.

*Figure 3b, what are the concentrations used in normal cells? Show the killing curve with at least 5 concentrations (covering the range used for malignant cells) and calculate AUC to compare.

We would like to add that the concentration range of 1nM- 10 µM was used for both cancer cells and normal cells.

The authors stated the concentrations used however did not present the killing curve of these compounds on normal cells. The concentration used in Figure 3b is still missing. Therefore, the claim regarding selectivity over normal cells is invalid.

*Figure 3e, compensation problem or cytometer setup problem. False claim.

The authors apologize for the ambiguity. We would like to humbly state that the experiments were carried out on BD Flow cytometer which has an auto-compensation with in default settings. Additionally, as suggested by the expert reviewer, we performed manual compensation for the fcs files and the plots are now provided in the revised Figure 3e.

The separation still looks awful. The reviewer suspect voltage was not setup correctly. Also, these cells were analyzed by AnnexinV-APC+PI (not mentioned in method section) where other parts the paper AnnexinV-AF488 is used. Results from different assays cannot be compared side by side. Please explain the inconsistency. Also, it is unclear which compound is used.

*Figure S4b, in the illustration of Annexin V-PI staining, compensation is clearly off as quite a few cells showed PE positive(Q2) in Annexin single staining. PI is a strong stain in the PE channel yet the flow plots showed intensity between 10^2 to 10^3, suggesting the voltage was not set correctly. Check the flow plot out in the manufacturer's guidebook(https://www.thermofisher.com/document-connect/document-connect.html?url=https://assets.thermofisher.com/TFS-Assets%2FLSG%2Fmanuals%2Fmp13241.pdf). 

The authors thank the reviewer for this comment. In this study, Annexin V staining was performed after 4 hours of treatment with the compounds. This was done to capture the cells at the early apoptotic phase and check the efficacy of the compounds in triggering Apoptosis. Further, we would like to humbly state that the experiment was carried out according to the manufacturer’s instructions. Untreated unstained and untreated single stained (only Annexin V and only PI) were used for gating and compensation. We sincerely hope this clarifies the data presented.

The authors did not address the reviewer’s concern. In Annexin V only stain, there are PI positive cells which indicates a compensation error. Authors did not address the voltage concern. S4d, another invalid claim because counting beads is not used to calculate total cells acquired. The number of cells acquired can differ by flow speed/time etc.

*Figure S4c, why the authors stained Annexin V and PI separately and how could it be possible to estimate double positive cells as they stated in line 218?

We apologize for the ambiguity in expressing the results. We would like to state that the staining wasn’t done separately but the population distribution of Annexin and PI positive cells in the sample were plotted separately. The Annexin V apoptosis assay was done according to the manufacturer’s protocol. Further, the readout for our experiment is to capture the cells undergoing early apoptosis, that is, when PI stains and also allows Annexin to enter the cells and additionally stain the cytosolic side of the plasma membrane. Therefore, Annexin PI stained (double positive) apoptotic cell populations always show a population which is high in Annexin and low in PI as presented in the figure. SM396 showed increased level of Annexin V positive cells when compared to other treated samples (highlighted in the MS).

The authors now have move single color stains to S4b. Authors does not seem to understand Annexin-PI staining. Early apoptosis cells are defined as AnnexinV+ PI- instead of double positive (PMC3169266). The reviewer understands circled population in FigureS4a is double positive, yet it does not change the fact that AnnexinV compensation is off.

*Figure 3d, indicate how S phase is calculated, and show the flow plots in addition to statistics in Sup

The authors did not address the reviewer’s concern.

*Fig2d. What does the right flow panel indicate? How could one even overlay two flow plot side by side on the same x and y axles?

We apologize for the ambiguity in representing the results. We have now revised all the information in the figure, corrected the merged axes and is presented in Figure3D

The authors have corrected the figure panel. However, y axis label and cell phase labels are still missing.

*Figure 3f, vehicle control showed >50% apoptotic cells, suggesting something is significantly wrong with this experiment, eg. contamination, wrong culture condition, etc.

We agree with reviewer, this could be possible due to our vehicle control concentration (10mM) used for the assay. We anticipate this could a probable reason for the observed result. However, we present the normalized apoptotic rate from control and treated samples to show the effect of the compounds. Kindly refer to following paper for more details on apoptotic rate PMID: 12210142.

It is unclear what “vehicle” control was used and why it was used at such high concentration. Apoptotic rate is used to include fragmented mitotic bodies that would be eliminated by washing or missed on a cytometer. AR= (number of AnnexinV+ cells+ number of fragmented cells)/total cell seeded. While AR can be calculated in this 6-hour treatment (no proliferation), the number of cells requires absolute cell counting by beads. The reviewer does not believe the authors did any absolute cell counting by beads in any of their experiments since it wasn’t mentioned. As a result, the AR presented lacks credibility. Regardless of which method is used to describe apoptosis, the fact that vehicle control reached IC50 stands, which significantly weakens the impressions(potent) the authors wanted to convey.

Line 211. Annexin V+ ve?

The authors apologize for this error, the line refers to BRdU assay, in which the 7AAD fraction of the samples were discussed.

The reviewer’s concern is addressed.

Line 223. Vis.,???

As suggested by the expert reviewer, we have now modified the sentence.

The reviewer’s concern is addressed.

*Line 221-249. The authors claim the apoptosis these three agents induced is BCL-2 dependent based on their observation of cytochrome C release and membrane depolarization. Of note, these two assays could only indicate apoptosis happened. KO/KD experiments and rescues are needed to verify BCL-2 dependency. 

The authors thank the insightful comment of the expert reviewer and agree the need for KO/KD experiments. We are planning additional experiments as future work for this project.

Please do not make any claim that these agents induced apoptosis is BCL-2 dependent for example Line 499. SM216, SM396, and SM949 induce BCL-2 dependent intrinsic apoptotic pathway.

Figure 5a, please indicate how many plates you did. In addition, the concentration used and the rationale for using that concentration.

The authors thank the expert reviewer for this observation. In this study, we used duplicates for each concentration and the experiments were repeated twice (n=2), we choose the IC50 values calculated from the MTT assays for each compound as treatment concentration for all the assays to test the efficacy of the compounds.

Please show the effect at all concentrations. IC50 may not be reached every time.

Line 290, human blood cells is vague and inaccurate. RBC only

The authors apologize for this ambiguity. The sentence was rephrased as suggested. (line290)

The reviewer’s concern is addressed.

Figure 6. in vivo studies, please explain why these compounds were given through i.p.

The authors thank the reviewer for this important note. The xenograft model is a robust model and is well suited for proof of concept study. The investigational compounds were injected through the IP route for greater drug absorption, ease of administration and importantly less trauma to the experimental animals.

The reviewer’s concern is addressed.

*Figure 6e bottom. a scale by the tumor is required to visualize tumor mass.

The authors apologize for this oversight, however we would like to state that we measured the tumors using a digital Vernier Calipers and wet weights of excised tumors using a weighing balance. We apologize for not using a scale while taking the photographs. We ensure to use it in future experiments.

Without a scale with photographs, these pictures can be misleading and should be removed because the distance between the tumor and the camera will significantly impact visual size.

*Figure 6d, authors need to show the section area staining by H&E, PCNA, and caspase3. Otherwise, there is no way to compare them.

We thank the reviewer for the comment and we have included the H&E stained photomicrographs in the panel.

In addition to H&E staining. It is critical and almost a common sense thing that researchers need to stain consecutive slides(H&E, markers) so people can visualize the same area of tissue and interpret the staining. The staining pictures are from different areas of a slide and therefore cannot be compared. 

In the introduction, please discuss the function of BH4 domain and its binding proteins and function beyond BH3 proteins.

We thank the reviewer for the comment we have now included the function of BH4 and its potent interactions in the introduction.

The reviewer’s concern is addressed.

Author Response

We thank the reviewer for the constructive comments. 

Answers to the comments of Reviewer 3

  1. The authors added a dose-repone curve of top three compounds. However, some compounds are not potent to reach 50% viability (IC50) eg. MCF7+SM396 and MDA+SM216 treatment combinations. Therefore, the authors must have imputed the IC50 based on the dose curve, which is inaccurate and misleading. Also, Fig S5 needs modifications (inconsistent front&size, style, and compound name is missing). The authors stated they calculated AUC yet results are not included.

The data was included in the supplementary figure S5, please find the revised figure. We would like to kindly mention that the compounds have reached 50% inhibition for the above mentioned conditions, in order to give a clear depiction, we showed only the datapoints to clarify the ambiguity. We have not imputed the IC50 values. Please refer to the supplementary figure S5.

  1. The authors stated the concentrations used however did not present the killing curve of these compounds on normal cells. The concentration used in Figure 3b is still missing. Therefore, the claim regarding selectivity over normal cells is invalid.

We apologize for the oversight, the concentrations were now included in the figure 3b, the dose response curves for the normal cells are included as supplementary figure S5b.

  1. The separation still looks awful. The reviewer suspect voltage was not setup correctly. Also, these cells were analyzed by AnnexinV-APC+PI (not mentioned in method section) where other parts the paper AnnexinV-AF488 is used. Results from different assays cannot be compared side by side. Please explain the inconsistency. Also, it is unclear which compound is used.

We apologize for the confusion, we used Annexin AF-488 for the Annexin assay. Please find the revised figure and the representative image treated sample

  1. The authors did not address the reviewer’s concern. In Annexin V only stain, there are PI positive cells which indicates a compensation error. Authors did not address the voltage concern. S4d, another invalid claim because counting beads is not used to calculate total cells acquired. The number of cells acquired can differ by flow speed/time etc.

We kindly request the reviewer to note that the flow plot showed no PI+ve cells in the Annexin only stained condition(Figure 3D). The voltage conditions were set with cells, accordingly. However, the voltages were set at the range of 102 for negatives and 105 for positives.

  1. The authors now have move single color stains to S4b. Authors does not seem to understand Annexin-PI staining. Early apoptosis cells are defined as AnnexinV+ PI- instead of double positive (PMC3169266). The reviewer understands circled population in FigureS4a is double positive, yet it does not change the fact that AnnexinV compensation is off.

The percentage Annexin positive cells were calculated from Annexin+ PI-, only Annexin + in double positive population and the results are presented in the Figure 3F.

  1. *Figure 3d, indicate how S phase is calculated, and show the flow plots in addition to statistics in Sup

The S-phase population was calculated using the cell cycle module of FlowJo software. The flow plots and additional statistics are presented as supplementary figure S6.

  1. *Fig2d. What does the right flow panel indicate? How could one even overlay two flow plot side by side on the same x and y axles?

We apologize for the ambiguity in representing the results. We have now revised all the information in the figure, corrected the merged axes and is presented in Figure 3D

  1. The authors have corrected the figure panel. However, y axis label and cell phase labels are still missing.

We have included the cell phase labels, please refer to revised figure 3D.

  1. It is unclear what “vehicle” control was used and why it was used at such high concentration. Apoptotic rate is used to include fragmented mitotic bodies that would be eliminated by washing or missed on a cytometer. AR= (number of AnnexinV+ cells+ number of fragmented cells)/total cell seeded. While AR can be calculated in this 6-hour treatment (no proliferation), the number of cells requires absolute cell counting by beads. The reviewer does not believe the authors did any absolute cell counting by beads in any of their experiments since it wasn’t mentioned. As a result, the AR presented lacks credibility. Regardless of which method is used to describe apoptosis, the fact that vehicle control reached IC50 stands, which significantly weakens the impressions(potent) the authors wanted to convey.

DMSO was used as vehicle control. The reason for using 10mM stock is to solubilize the compounds  as recommended. The manual compensation on the treated samples with respect to unstained and single stained cells gave <20% of apoptotic cells in the control sample

  1. Please do not make any claim that these agents induced apoptosis is BCL-2 dependent for example Line 499. SM216, SM396, and SM949 induce BCL-2 dependent intrinsic apoptotic pathway.

We have rephrased the heading as suggested. But, we presume that the results of two key experiments, BH3 profiling and Cyt C ELISA should imply the induction of membrane depolarization and Cyt c release upon treatment which are reported to be BCL-2 dependent events.

  1. Please show the effect at all concentrations. IC50 may not be reached every time.

The assay was performed only at the IC50 concentrations which were consistent for all the assays on MDA-MB-231 cells. We confirmed the activity of compounds multiple times on MDA-MB-231 cells and the IC50 was consistent.

  1. *Figure 6e bottom. a scale by the tumor is required to visualize tumor mass.

The authors apologize for this oversight, however we would like to state that we measured the tumors using a digital Vernier Calipers and wet weights of excised tumors using a weighing balance. We apologize for not using a scale while taking the photographs. We ensure to use it in future experiments.

  1. Without a scale with photographs, these pictures can be misleading and should be removed because the distance between the tumor and the camera will significantly impact visual size.

As suggested we have removed the tumor pictures from the panel.

  1. In addition to H&E staining. It is critical and almost a common sense thing that researchers need to stain consecutive slides(H&E, markers) so people can visualize the same area of tissue and interpret the staining. The staining pictures are from different areas of a slide and therefore cannot be compared. 

We thank the reviewer for the suggestion and kindly refer the revised section images in Figure 6D.